# OpenReview forum: "PII-Scope: A Comprehensive Study on PII Extraction Attacks in LLMs"
_TMLR — Rejected by TMLR_

### Review · Reviewer_jpwK · 2025-03-04

**Summary Of Contributions:**

This paper proposes a taxonomy of PII attacks and examines the performance of these attacks, as well as their sensitivity to internal attack hyperparameters. It highlights the high performance of PII attacks in two realistic scenarios: one where repeated or diverse input prompts are used to query the LLM multiple times, and another where previously extracted PIIs are iteratively leveraged to enhance subsequent extractions.

**Audience:**

Yes

**Claims And Evidence:**

No

**Requested Changes:**

1. Writing:
There are several areas that need improvement.

For instance, the reasoning behind the title "PII-Scope: A Benchmark for Training Data PII Leakage Assessment in LLMs" is unclear to me. The paper seems more like a study rather than a benchmark.

Additionally, the methodology name "PII-Scope" is never mentioned again after the introduction.

The organization of the chapters is also confusing. The taxonomy section is followed directly by the experiments, making it difficult to identify the paper's key contribution.

Furthermore, the abbreviation "PII" should be introduced in the abstract rather than in the introduction.

On page 5, the remark regarding the SPT Attack is distracting. Soft-prompt tuning has been around for a long time and can also be applied to PII attacks, so there's no need to emphasize this in such a distracting manner.

I suggest ensuring consistency between the statements of contribution in the abstract and the introduction. For instance, the abstract mentions that PII extraction rates can increase by up to fivefold when targeting the pretrained model, while the introduction states that extraction rates can improve by up to threefold with a limited query budget. It is important to align these statements to avoid confusion. This makes it hard to follow.

2. Experiments Scope:

I suggest conducting more experiments beyond the current dataset and models.

There are numerous types of PII, so it is important to explore beyond just names and emails. This broader exploration is crucial for the study, as different types of PII may exhibit varying performance characteristics.

**Strengths And Weaknesses:**

Strengths: The study covers relatively comprehensive types of PII attacks.

Weaknesses:
1, The writing needs to be significantly improved.
2, The experiment settings are not diverse enough.
3, The contribution of this paper is not clear to me. The title is confusing.

---

> ### Author Response · Authors · 2025-04-07
>
> ### **Experimental Scope**
> - We greatly appreciate the reviewer’s insightful comments regarding the experimental scope. In response to these, we have conducted additional experiments to further strengthen the generalizability of our findings.
> - Specifically, we added **Section 10** to evaluate email PII extraction attacks on the Pythia 6.9B model in both pretrained and finetuned settings. Furthermore, in **Section 12**, we studied the leakage rates of the finetuned **LLaMa7B** model.
> - We also added **Section 11** with experiments on **phone number** PII extraction attacks on both GPT-J-6B and Pythia 6.9B in pretrained and finetuned settings.
> - Our additional experimental results further validate our prior findings, confirming that extraction rates significantly improve with repeated querying and in continual settings.
>
> ### **Writing**
> - Following the reviewer’s suggestions, we have updated the paper title to: "PII-Scope: A Comprehensive Study on  PII Extraction Attacks in LLMs" to better reflect that our work focuses on a deeper study of PII attacks rather than simply providing a benchmark dataset.
> - Additionally, to better highlight our contributions, we have added **Section 3**, which provides an overview of PII-Scope, the core components of the current framework, and the key takeaways from our extensive experiments.
> - We have removed the remark discussing the broad application of SPT attacks, as we agree that it could detract from the main focus of the discussion.
>
> We believe that these clarifications address the reviewer’s concerns and further strengthen the generalizability of our findings related to increased extraction rates with advanced attacker capabilities.

---

### Review · Reviewer_bR9W · 2025-03-13

**Summary Of Contributions:**

This submission introduces PII-Scope, a comprehensive benchmark for evaluating the effectiveness of personally identifiable information (PII) extraction attacks on LLMs across various threat settings.

The authors (1) present a taxonomy of PII attacks categorized by threat models (white-box or black-box) and data accessibility assumptions, and (2) systematically analyze the sensitivity of different PII attacks to their internal hyper-parameters. Their findings reveal that template structures closely resembling the original data yield significantly better extraction, ICL attacks are more influenced by the quality of selected demonstrations, and SPT attacks are highly sensitive to prompt initialization.

They further (3) examine under more realistic attack scenarios, such as multi-query setting, demonstrating that PII extraction rates can significantly increase with advanced attacks.

**Audience:**

Yes

**Broader Impact Concerns:**

The findings of the submission may be exploited to enhance the performance of PII extraction attacks in LLMs, posing privacy risks to individuals, hence, a broader impact statement is encouraged to be added.

**Claims And Evidence:**

Yes

**Requested Changes:**

- In Section 5.4, Sufficient contextual information exists with $L=25$ tokens, (1) what if to further examine the performance when $L<25$ tokens? (2) Does the content itself matter? For instance, "from John" and "John" may provide the same contextual information while the length is different.

- In Figure 8 (b), Random tokens achieve better performance than Task-aware tokens, although the authors mentioned in the submission that they are unable to hypothesize the exact reason for this, would it be possible that overfitting happens with Task-aware tokens?

- In Figure 11 (a), Synthetic data (altering only the name) achieves better performance than real data with template C, could the authors provide some explanation about this behaviour?

**Strengths And Weaknesses:**

**Strengths**

- The submission presents a well-structured benchmark PII-Scope that systematically evaluates PII extraction attacks and analyzes the sensitivity to internal hyper-parameters, offering empirical insights into the factors that affect the extraction performance.

- It also goes beyond single-query attacks, exploring more advanced scenarios and revealing that the effectiveness of PII extraction attacks has been significantly underestimated.

- The benchmark relies on filtered Enron PII datasets, which reduce the bias risks.

**Weaknesses**

- The submission mainly focuses on GPT-J-6B (as stated in the Limitations as well), which may not fully capture the diversity of LLM architectures. Expanding to more diverse LLMs could help with the findings.

- The submission does not explore any potential mitigation strategies. For example, DP (differential privacy) and model compression are common methods against data reconstruction attacks, which may help defend against PII attacks in LLMs.

---

> ### Author Response · Authors · 2025-04-07
>
> We would like to thank the reviewer for their valuable feedback. Below, we address the specific concerns raised and provide clarifications based on your insightful suggestions.
>
> #### 1. **Expanding to More Diverse LLM Architectures:**
> We expand our experiments with more diverse LLM architectures to  strengthen the generalizability of our findings.
> - We have added **Section 10**, where we evaluate email PII extraction attacks on the Pythia 6.9B model in both pretrained and finetuned settings.
> - Additionally, **Section 11** now includes experiments on phone number PII extraction attacks using both GPT-J-6B and Pythia 6.9B in pretrained and finetuned settings.
> - Lastly, in **Section 12**, we present experiments with the finetuned **LLaMa-7B** model in standard settings, benchmarked against a scrubbing-based defense.
> These additions further enrich our study and improve the robustness of our conclusions, and we’re excited about the enhanced breadth this brings to our analysis.
> #### 2. **Mitigation Strategies:**
> - Pretrained base LLMs are typically trained on large corpora of publicly available data, such as web-crawled content, to predict the next token. At the scale of pretraining, most recent LLMs rely on PII filtering techniques, which are generally automated. In **Section 11**, we present results using a PII-scrubbing-based defense on the finetuned **LLaMa-7B** model, where we show that our findings of increased leakage rates with repeated querying also apply to models defended with scrubbing.
> #### 3. **Contextual Information in Section 5.4:**
> We truly appreciate the reviewer’s thoughtful question about the impact of contextual information on PII extraction.
> - **Performance with Smaller Context Lengths:** In our experiments, we focused on the effect of token length on PII extraction performance, using context lengths of 25, 50, and 100 tokens. Upon reviewer's suggestion, we experimented with even shorter context lengths (5 and 10 tokens) also yield relatively high extraction rates: **6.81%** and **7.14%**, respectively, compared to **3.9%** with no context. For comparison, the highest extraction rates with 25, 50, and 100 context tokens were **8.4%**, **8.1%**, and **8.8%**, respectively.
> - **Does Content Itself Matter?**  We believe that for attacks such as ICL & PII-Compass, the content in the context prompt plays a more significant role than the length of the context from our experiments. For example, a 5-token prefix (**'com'; \n> '**) can improve the extraction rate of the template D prompt from **3.9%** to **6.8%**, while a slightly different but semantically closer 5-token context (**".com>,\t<"**) improves the extraction rate to **5.5%**. Moreover, a longer 10-token context (e.g., **"stamper@omm.com>,\t<"**) results in an extraction rate of only **4.8%** for template D. These prelimilimary results suggest that the factors affecting PII extraction are multifaceted  and deeply tied to the specific context that triggers the PII-generation. We have updated Figure 7(c) and elaborated on this in Section 6.4. Additionally, we have made minor updates to Section 13 to highlight how these findings can guide future mechanistic interpretations of attack success.

---

> ### Author Response · Authors · 2025-04-08
>
> #### 4. **Performance of Random Tokens vs. Task-Aware Tokens in Figure 8(b):**
> We greatly appreciate the reviewer’s observation regarding the discrepancy in performance between random tokens and task-aware tokens in **Figure 8(b)**. As we mentioned in the submission, we were unable to hypothesize the exact cause of this behavior is due to isolated case of overfitting.  To understand the reasons for this phenemenon, we believe further exploration is needed into how and why a given context (such as soft prompt initialization) influences PII extraction. This requires a mechanistic exploration of how prompts (both soft prompts here and hard prompts in the ICL and PII-Compass) trigger the PII-sensitive neurons across the layers of LLMs. We hope our work will inspire further research on PII memorization in LLMs, studying **how and why** different input prompts with different attacks influence the PII extraction.
> #### 5. **Synthetic Data vs. Real Data in Figure 11(a):**
> We thank the reviewer for pointing out this result in **Figure 11(a)**, where synthetic data (altering only the name) achieves slightly better performance **on average** than real data with template C. We acknowledge that we have not yet fully isolated the reasons for this behavior. However, we observe that the highest extraction rates with real data are higher than the highest extraction rates with synthetic data (altering only the name) for template C. Additionally, the extraction rates with real data exhibit significant variance, suggesting the need for a deeper mechanistic understanding triggered by different prompt constructions, whether crafted with various attacks or using synthetic datasets.
> While we are not able to precisely identify the reasons for this empirical phenomenon, we believe that our findings, across different attacks, can aid in the mechanistic interpretation of LLM behavior and further contribute to the development of stronger attacks. We have added discussion to this in Section 13 as an interesting direction for future work.
>
> #### **Broader Impact Concerns**
> Thank you for suggestion. We have added a section on broader impact concerns to better position the impact of our work.

---

### Review · Reviewer_3Y8o · 2025-03-24

**Summary Of Contributions:**

This paper proposes a benchmark of PII for LLMs that considers different threat settings. The authors first taxonomy existing PII attacks against LLM and categorize them based on different threat models. Based on a newly-composed dataset, this work mainly studies hyperparameters that are crucial to PII effectiveness, including repeated and diverse querying. One finding is that single-query attacks underestimate PII leakages, and another is that finetuned models are more vulnerable than pre-trained models. This work aims to provide an empirical benchmark for PII extraction attacks.

**Audience:**

Yes

**Broader Impact Concerns:**

None.

**Claims And Evidence:**

No

**Requested Changes:**

I see this paper still needs some effort as a benchmark paper.
Please clarify the generalization of the less-biased dataset, especially regarding this work as a benchmark.
Please discuss the overall methodology's generalization and clarify the benchmark design's core.
Regarding the limited dataset and models, please provide some explanations.

**Strengths And Weaknesses:**

Strengths:

This paper is well organized, where research questions and backgrounds are clearly stated, which is very important for a benchmark paper. Different attacks are also well categorized and discussed.

Weaknesses:

One core contribution of this work is to provide a new balanced benchmark dataset where different domains are balanced. I agree with the authors that bias is indeed an issue that makes the PII evaluation obscured, leading to highly-overlapped domains. However, I have the following two concerns related to each other. One concern is that the domain biases are naturally embedded in the dataset, resembling the real-world nature of the distribution. Tweaking the training data might be helpful, but I would keep the distribution of the test data the same as the raw data to provide a practical evaluation. For example, there are indeed more hotmails in practice. Getting rid of them in either D_adv or D_eval makes less sense since there is a huge chance it is included in practice.  Another concern is that limiting the domain may limit the adversaries' capabilities, providing a misleading benchmark regarding practical PII risks. For example, as the authors also mentioned in Figure 8, there might be an overfitting. Would it work better on hotmails if the original distribution is not tweaked? If so, is the PII-Score benchmark underestimate the PII leakage for hotmails?

The multi-query attack is not surprising. For example, a few-shot setting has also been studied by Huang et al., 2022a. The authors definitely provide a comprehensive study showing that single-query PII attacks are limited. However, regarding the related work, this finding is not surprising. I would suggest the authors to provide a deeper analysis regarding the existing works based on your solid analysis.

The authors select GPT-J-6B as the baseline model. As a benchmark paper, I would suggest that the authors could introduce more baselines and show the generalization of the provided knowledge.

Minor:

Gemma -> Gemini?

Table 1. True-prefixes are true prefixes are used interchangeably.
The order of black-box and white-box setups can be made the same.

---

> ### Author Response · Authors · 2025-04-07
>
> We are grateful for the reviewer’s valuable insights, which have helped us refine our approach and improve the clarity of our findings.
>
>
> #### **Generalization of the Less-Biased Dataset**
> - We agree with the reviewer that email domain biases are naturally embedded in real-world data, and our current balanced dataset might underestimate the actual leakage. We emphasize that we carefully selected data subjects without email domain overlap between $D_{adv}$ and $D_{eval}$ to ensure our analysis remains free from dataset bias. This choice strengthens the robustness of our findings and helps establish a baseline for understanding the impact of privacy leakage. However, we recognize that real-world scenarios, particularly for email PII, often involve domain biases that attackers can leverage to launch attacks. While we limit the attacker's capability under the assumption of no domain overlap with evaluation data subjects, we have also explored advanced attack capabilities in multi-query and continual settings to more accurately reflect real-world attack scenarios.
>
> - To further generalize our findings beyond email PII, we have expanded our experiments to include phone number PII, focusing on the extraction of exact 10-digit phone numbers. This extension provides additional context for understanding PII leakage across different types of PII, while avoiding the issue of email domain contamination seen in email PII.
>
> #### **Clarification of Benchmark Design’s Core**
>
> - The core of our benchmark design is to quantify privacy leakage from PII extraction attacks under various query settings, while also assessing the effectiveness of repeated and continual querying with advanced attacker capabilities. This framework allows us to evaluate how well LLMs resist these types of attacks. Ultimately, the goal of our study is to provide a reliable evaluation framework that can be adapted to different LLMs and threat configurations. This benchmark study aims to serve as a valuable tool for the broader community, enabling them to assess and compare privacy risks across various threat models and attack strategies, while advocating for adversarial evaluations of privacy leakage in LLMs. Additionally, to better highlight our contributions, we have added Section 3, which provides an overview of PII-Scope, the core components of the current framework, and the key takeaways from our extensive experiments.
>
>
> #### **Explanation of the Limited Dataset and Models**
>
> - We appreciate the reviewer’s concern regarding the limited number of models and datasets used in this study. In response, we have expanded our analysis to include additional models and PIIs, incorporating the reviewer’s suggestion to better generalize our findings across various configurations.
> - Specifically, we have added **Section 10** to evaluate email PII extraction attacks on the Pythia 6.9B model in both pretrained and finetuned settings.
> - Additionally, **Section 11** now includes experiments on phone number PII extraction attacks using both GPT-J-6B and Pythia 6.9B in pretrained and finetuned settings.
> - Finally, **Section 12** presents experiments with the finetuned LLaMa-7B model in standard settings, benchmarked against a scrubbing-based defense.
>
> #### **Related Work**
>
> - We appreciate the reviewer acknowledging our comprehensive analysis of the limitations of single-query PII attacks. However, we would like to clarify that Huang et al., 2022a focused on zero to five-shot attacks with a static demonstration prompt in a single-query setting. While the findings in a multi-query setting may not be surprising, our contribution lies in demonstrating the substantial variance in extraction rates for a given $k$-shot setting (see Figure 7(b)). Specifically, we show that even a 2-shot demonstration matches the performance of a 32-shot demonstration in a single-query setting. Furthermore, we investigate advanced attack capabilities, particularly in multi-query settings and in continual settings, which have not been studied in the context of PII leakage in LLMs. Our findings reveal significantly higher extraction rates under these more advanced attack strategies. We hope this clarification helps to emphasize the novelty and significance of our work in relation to existing studies.
>
>
> We hope that these clarifications address your concerns and further enhance the clarity and generalisation of our findings.

---

### Author Response · Authors · 2025-04-07
**Revision**

We thank the reviewers for their insightful comments, which have been invaluable in refining our work. We have incorporated the suggested changes, particularly the additional experiments and clarifications, into the latest version of the PDF. The following revisions have been made to the submission:

- **Section 3** now discusses an overview of the PII-Scope framework and outlines the presentation of our work.
- **Section 10** includes additional results on the Pythia 6.9B model for email PII extraction.
- **Section 11** presents new results on phone number PII extraction for both the original GPT-J-6B model and the newly evaluated Pythia 6.9B model.
- **Section 12** provides results using the finetuned LLaMa7B model, benchmarked with a PII-Scrubbing defense.

We hope these revisions address the reviewers' concerns and further clarify our contributions.

---

> ### Author Response · Authors · 2025-05-01
> **Follow up**
>
> Dear Reviewers,
>
> We would be grateful if the you could take a look at our responses and revision. Please let us know if you have more questions.
>
> Sincerely,
>
> Authors

---

### Decision · Action_Editor_Kt39 · 2025-05-09

**Recommendation:** Reject

**Comment:**

In this submission, the authors proposed to benchmark the state-of-the-art methodologies for personally identifiable information (PII) extraction attacks based on the LLMs with various threat models and data accessibility assumptions.  While all reviewers acknowledge the comprehensive nature of the submission in evaluating different types of PII attacks, they still have several concerns after the revisions:

- The generalization of the benchmark is limited.

- The number of datasets and LLMs included in the benchmark is insufficient.

- The clarity and quality of writing could be improved.

The AE has reviewed the submission and agrees with these assessments. Given that the primary contribution of this work is the benchmark, it still requires significant enhancements to meet expectations.

**Audience:**

Yes.

**Claims And Evidence:**

No.

One reviewer pointed out the experimental scope is limited, and another reviewer pointed out the generalization issue of the proposed benchmark is not well discussed.

**Resubmission Of Major Revision:**

The authors may consider submitting a major revision at a later time.